# Circulatory Rejuvenated EPCs Derived from PAOD Patients Treated by CD34^+^ Cells and Hyperbaric Oxygen Therapy Salvaged the Nude Mouse Limb against Critical Ischemia

**DOI:** 10.3390/ijms21217887

**Published:** 2020-10-23

**Authors:** Yin-Chia Chen, Jiunn-Jye Sheu, John Y. Chiang, Pei-Lin Shao, Shun-Cheng Wu, Pei-Hsun Sung, Yi-Chen Li, Yi-Ling Chen, Tien-Hung Huang, Kuan-Hung Chen, Hon-Kan Yip

**Affiliations:** 1Division of Thoracic and Cardiovascular Surgery, Department of Surgery, Kaohsiung Chang Gung Memorial Hospital and Chang Gung University College of Medicine, Kaohsiung 83301, Taiwan; w780726@cgmh.org.tw (Y.-C.C.); cvsjjs@gmail.com (J.-J.S.); 2Institute for Translational Research in Biomedicine, Kaohsiung Chang Gung Memorial Hospital, Kaohsiung 83301, Taiwan; e12281@cgmh.org.tw (P.-H.S.); rylchen.msu@gmail.com (Y.-L.C.); tienhunghuang@gmail.com (T.-H.H.); 3Center for Shockwave Medicine and Tissue Engineering, Kaohsiung Chang Gung Memorial Hospital, Kaohsiung 83301, Taiwan; 4Department of Computer Science and Engineering, National Sun Yat-Sen University, Kaohsiung 80424, Taiwan; chiang@cse.nsysu.edu.tw; 5Department of Healthcare Administration and Medical Informatics, Kaohsiung Medical University, Kaohsiung 80756, Taiwan; 6Department of Nursing, Asia University, Taichung 41354, Taiwan; m8951016@gmail.com; 7Regenerative Medicine and Cell Therapy Research Center, Kaohsiung Medical University, Kaohsiung 80756, Taiwan; shunchengwu@hotmail.com; 8Orthopaedic Research Center, Kaohsiung Medical University, Kaohsiung 80756, Taiwan; 9Post-Baccalaureate Program in Nursing, Asia University, Taichung 41354, Taiwan; 10Department of Cardiology, Department of Internal Medicine, Kaohsiung Chang Gung Memorial Hospital and Chang Gung University College of Medicine, Kaohsiung 83301, Taiwan; ryichenli@gmail.com; 11Department of Anesthesiology, Kaohsiung Chang Gung Memorial Hospital and Chang Gung University College of Medicine, Kaohsiung 83301, Taiwan; amigofx35@gmail.com; 12Department of Medical Research, China Medical University Hospital, China Medical University, Taichung 40402, Taiwan; 13Division of Cardiology, Department of Internal Medicine, Xiamen Chang Gung Hospital, Xiamen 361028, China

**Keywords:** critical limb ischemia, endothelial progenitor cells, nude mice, angiogenesis, hyperbaric oxygen therapy

## Abstract

This study tested whether circulatory endothelial progenitor cells (EPCs) derived from peripheral arterial occlusive disease (PAOD) patients after receiving combined autologous CD34+ cell and hyperbaric oxygen (HBO) therapy (defined as rejuvenated EPCs) would salvage nude mouse limbs against critical limb ischemia (CLI). Adult-male nude mice (*n* = 40) were equally categorized into group 1 (sham-operated control), group 2 (CLI), group 3 (CLI-EPCs (6 × 10^5^) derived from PAOD patient’s circulatory blood prior to CD34^+^ cell and HBO treatment (EPC^Pr-T^) by intramuscular injection at 3 h after CLI induction) and group 4 (CLI-EPCs (6 × 10^5^) derived from PAOD patient’s circulatory blood after CD34^+^ cell and HBO treatment (EPC^Af-T^) by the identical injection method). By 2, 7 and 14 days after the CLI procedure, the ischemic to normal blood flow (INBF) ratio was highest in group 1, lowest in group 2 and significantly lower in group 4 than in group 3 (*p* < 0.0001). The protein levels of endothelial functional integrity (CD31/von Willebrand factor (vWF)/endothelial nitric-oxide synthase (eNOS)) expressed a similar pattern to that of INBF. In contrast, apoptotic/mitochondrial-damaged (mitochondrial-Bax/caspase-3/PARP/cytosolic-cytochrome-C) biomarkers and fibrosis (Smad3/TGF-ß) exhibited an opposite pattern, whereas the protein expressions of anti-fibrosis (Smad1/5 and BMP-2) and mitochondrial integrity (mitochondrial-cytochrome-C) showed an identical pattern of INBF (all *p* < 0.0001). The protein expressions of angiogenesis biomarkers (VEGF/SDF-1α/HIF-1α) were progressively increased from groups 1 to 3 (all *p* < 0.0010). The number of small vessels and endothelial cell surface markers (CD31^+^/vWF^+^) in the CLI area displayed an identical pattern of INBF (all *p* < 0.0001). CLI automatic amputation was higher in group 2 than in other groups (all *p* < 0.001). In conclusion, EPCs from HBO-C34+ cell therapy significantly restored the blood flow and salvaged the CLI in nude mice.

## 1. Introduction

Peripheral arterial occlusive disease (PAOD), a high prevalence aging-associated chronic disease, not only incurs huge public healthcare costs but also causes unacceptably high morbidity and mortality worldwide. PAOD, one of the major manifestations of systemic atherosclerosis [1], has been established to affect 12% of the adult population and up to 20% of the elderly [2]. Patients with PAOD may develop critical limb ischemia (CLI) at the late stage of the disease process [2,3]. Undoubtedly, the CLI commonly occurs when arterial blood flow is greatly restricted, resulting in perfusion in capillary beds and inadequately sustained microvasculature. Ultimately, hypoxia and exhausted energy develop in the tissues and cells in the ischemic area [4,5]. Importantly, clinical studies have delineated that thousands of patients are asymptomatic prior to the development of CLI [6,7], which poses an obstacle to early diagnosis and early treatment for the purposes of slowing or abolishing disease progression and the development of unacceptable complications.

Treatment for the CLI, therefore, remains a formidable challenge to clinicians [8]. Without appropriate treatment, 5-year mortality of patients with asymptomatic PAOD is estimated for up to 19% of those diagnosed; and increases to 24% for patients with symptomatic PAOD [9]. Additionally, one-year mortality for CLI is identified for as many as 25% of those diagnosed [10]. Of particular importance is that patients with PAOD have a significantly elevated incidence of cardiovascular morbidity and mortality [9].

Failure in salvaging the critical limb can lead to limb loss and the high cost of patient care following amputation [11,12]. While surgical or endovascular revascularization is currently utilized for the treatment of CLI with an acceptable success rate [13,14,15], for those patients who are not candidates for surgical or endovascular intervention and those who failed the revascularization or bypass occlusion, the clinical outcomes remain dismal [13,14]. Accordingly, an alternative strategy for the treatment of CLI patients who are refractory to conventional therapy is urgently necessary.

Growing data have demonstrated that cell therapy effectively restored blood flow in the ischemic area, resulting in improved ischemic related organ dysfunction mainly through angiogenesis, neovascularization, anti-inflammation, immunomodulation and tissue regeneration in various disease entities [16,17,18]. Thus, cell therapy has emerged as an attractive modality for the treatment of ischemic heart and cerebral vascular diseases that has been reported to give promising results in animal model studies and clinical trials [16,17,18,19,20]. However, a majority of the clinical trials demonstrated that stem cell therapy, including those of autologous bone marrow-derived mononuclear cells or circulatory derived autologous endothelial progenitor cells (EPCs), did not offer additional benefits for salvaging the ischemic limb and improving the clinical outcomes [21,22]. This result could be mainly due to the coexisting serious atherosclerosis of small vessels [8,10] and severe dysfunctions of EPCs, capillary beds and microvasculature in this subgroup of patients [8]. Thus, improving the EPC function through culturing (i.e., rejuvenated EPCs) and increasing the permeability of microvasculature for EPC migration and homing in on the ischemic area could be an innovative approach for the treatment of CLI.

Hyperbaric oxygen (HBO) therapy is a traditional therapy for patients with ischemic PAOD [23]. It is proposed that the underlying mechanism of HBO therapy that is predominantly involved in improving ischemic PAOD is the increase of vascular wall permeability and production of hypoxia-inducible factor (HIF)-1α and stromal cell-derived factor (SDF)-1α that enhance the angiogenesis, circulatory EPCs level and blood flow in the ischemic area [24,25]. Unsatisfactorily, the overall limb salvage and progression of the ischemic process have not been significantly decreased in patients receiving HBO therapy [26]. This may be due to the diffusion of oxygen into the ischemic organ that creates a hyperbaric environment being extremely limited [27,28] in severe PAOD patients. Interestingly, our recent study demonstrates that HBO increases the number and function of circulatory EPCs [25]. This finding highlights that HBO may play an accessory role in: (1) promoting the mobilization of EPCs from bone marrow to the circulation; (2) enhancing the capillary/microvascular permeability; (3) augmenting the intrinsic and extrinsic EPCs crossing the vessel wall and homing in on the ischemic area; and (4) acting as a contributor for repairing endothelial functions in the ischemic area via increasing oxygen diffusion into the ischemia, increasing the likelihood of successfully salvaging the limbs of CLI patients.

Based on the aforementioned issues [16,17,18,19,20,21,22,23,24,25,26,27,28], we tested the hypothesis that treatment with EPCs derived from PAOD patients who received combined CD34^+^ cell and HBO therapy (defined as rejuvenated EPCs) would offer additional benefits than either one therapy alone for protecting limbs against the CLI procedure in nude mouse.

## 2. Results

### 2.1. Ischemic/Normal Blood Flow (INBF) Ratio Measured by Laser Doppler Scan at Days 2, 7 and 14 after Left Femoral Artery Ligated and Totally Removed and Automatic Amputation of Distal Critical Ischemic Limb

To elucidate whether intramuscular administration of EPCs which were derived from PAOD patients prior to and after receiving combined therapy of HBO and CD34^+^ cells would salvage the CLI animals, a laser Doppler scan was utilized for determining the INBF ratio in each group of animals. The result demonstrated that by days 2, 7 and 14 after the CLI procedure, the INBF was highest in group 1 (i.e., SC), lowest in group 2 (CLI only) and significantly lower in group 3 (CLI + EPCs which were derived from PAOD patient’s circulatory blood prior to receiving combined HBO and CD34^+^ cell treatment) than in group 4 (CLI + EPCs which were derived from PAOD patient’s circulatory blood after receiving HBO therapy and CD34^+^ cell treatment four times) (Figure 1A–O).

Additionally, by day 28 after the CLI procedure, we identified that the number of automatic amputations of distal ischemic limbs was significantly higher in groups 2 and 3 than in groups 1 and 4, but showed no significant difference between groups 2 and 3 or between 1 and 4, suggesting only rejuvenated EPCs effectively preserved the limb from the CLI procedure (Figure 1P).

### 2.2. The Protein Expressions of Endothelial Cell Functional Integrity in CLI Zone by Day 28 after CLI Procedure 

To assess the impact of EPC therapy on protecting the integrity of endothelial cell integrity, the Western blot analysis of a quadriceps specimen, which was harvested from the ischemic zone, was performed. The result showed that the protein expressions of CD31, von Willebrand factor (vWF) and endothelial nitric-oxide synthase (eNOS), three indices of endothelial cell integrity, were highest in group 1, lowest in group 2 and significantly higher in group 4 than in group 3 (Figure 2).

### 2.3. Protein Expressions of Angiogenesis in CLI Zone by Day 28 after CLI Procedure 

We further evaluated whether EPC therapy would enhance the angiogenesis in the CLI area by performing Western blot analysis for the typical angiogenesis biomarkers. The results showed that the protein expressions of CXCR4, SDF-1α, VEGF and HIF-1α, four indicators of angiogenesis, were significantly progressively increased from groups 1 to 4, suggesting an intrinsic response to ischemic stimulation that was augmented by EPC therapy (Figure 3).

### 2.4. Protein Expressions of Fibrotic and Antifibrotic Biomarkers in CLI Zone by Day 28 after CLI Procedure 

To test whether EPC therapy would attenuate the protein level of ischemia-related fibrosis in the quadriceps muscle, the Western blot analysis was performed. As we expected, the protein expressions of TGF-ß and Smad3, two indicators of fibrosis, were lowest in group 1, highest in group 2 and significantly lower in group 4 than in group 3 (Figure 4A,B). On the other hand, the protein expressions of Smad1/5 and BMP-2, two indicators of antifibrosis, displayed an opposite pattern of fibrosis among the four groups (Figure 4C,D).

### 2.5. Protein Expressions of Apoptotic, Mitochondrial-Damaged and Mitochondrial-Integrity Biomarkers in CLI Zone by Day 28 after CLI Procedure 

Next, for the assessment of apoptotic biomarkers, we also performed the Western blot analysis. The results demonstrated that the protein expressions of mitochondrial Bax (mit-Bax), cleaved caspase 3 (c-Csp3) and cleaved poly (ADP-ribose) polymerase (c-PARP), three indicators of apoptosis and cytosolic cytochrome C (cyt-Cyto-C) and indicators of mitochondrial damage, were highest in group 2, lowest in group 1 and significantly higher in group 3 than in group 4 (Figure 5A–D), whereas the protein expression of mitochondrial cytochrome C (mit-Cyto-C), an indicator of mitochondrial integrity, exhibited an opposite pattern of apoptosis among the four groups (Figure 5E).

### 2.6. Cellular Expressions of Endothelial Cell and EPC Surface Markers in CLI Zone by Day 28 after CLI Procedure 

To confirm the therapeutic impact of EPCs on the integrity of endothelial cells, we utilized the findings of immunofluorescent (IF) microscopy. The results exhibited that the cellular expressions of CD31 and vWF, two indicators of endothelial cell surface markers, were highest in group 1, lowest in group 2 and significantly higher in group 4 than in group 3 (Figure 6).

Additionally, we utilized the same method to further confirm the expressions of angiogenesis factors in the ischemic quadriceps muscle. The IF imaging study displayed that the cellular expressions of CXCR4 and SDF-1α, two indices of angiogenesis biomarkers, were progressively increased from groups 1 to 4, implicating an intrinsic response to ischemic stimulation that was upregulated by EPC therapy (Figure 7).

### 2.7. Small Vessel Density in CLI Zone by Day 28 after CLI Procedure

As expected, the number of small vessels (i.e., ≤25 µm) in the ischemic area of quadriceps was highest in group 1, lowest in group 2 and significantly higher in group 4 than in group 3 (Figure 8).

### 2.8. Time Courses of Circulatory Endothelial Progenitor Cells among the Groups 

To elucidate the serial changes of the circulating number of EPCs, flow cytometric analysis was performed for the peripheral blood mononuclear cells (PBMNCs). Time courses of flow cytometric analysis for identification of circulatory endothelial progenitor cells among the groups were showed in Appendix A. The result showed that by day 0, the circulating number of c-kit/CD31^+^, CD31/Sca-1+, KDR/CD34^+^ and VE-Cadherin/CD34^+^ cells, four indices of EPCs, did not differ among the four groups (Figure 9A–D). However, by day 4 and 15 after the CLI procedure, these parameters were progressively and significantly increased from groups 1 to 4, suggesting there was an intrinsic response to ischemic stimulation, which was further upregulated by EPC-HBO therapy (Figure 9E–L). An interesting finding was that the peak level of these circulating EPC biomarkers was reached by day 4 after the CLI procedure.

## 3. Discussion

This study investigated the impact of cultured EPCs, which were derived from the circulatory blood of severe PAOD patients, on CLI nude mice and yielded several striking implications. First, this study found that the number of automatic amputation of distal limbs did not differ between the CLI group and EPC^Pr-T^ group, suggesting that EPCs derived from severe PAOD patients prior to being rejuvenated did not provide benefits for salvaging the distal part of the ischemic limb in nude mice. Second, the INBF value only showed weak statistical significance in the EPC^Pr-T^ group as compared with CLI only. These findings implicated that the EPC function was remarkably impaired in those of severe PAOD patients prior to receiving rejuvenation therapy. Third, on the other hand, the number of automatic amputations of distal portion of ischemic limbs was substantially reduced, whereas the INBF value was remarkably increased in EPC^Af-T^ (i.e., rejuvenated EPCs) animals than in those of EPC^Pr-T^ animals, highlighting that only those of rejuvenated EPCs effectively preserved the limb from the CLI procedure.

EPC therapy has been demonstrated to effectively enhance angiogenesis, restore the blood flow in the ischemic zone and improve ischemic related organ dysfunction by abundant data [16,17,18,19,20]. However, when the published data were carefully examined, we found that severe PAOD patients usually responded poorly to the EPC therapy [21,22], resulting in failure to save severe ischemic limbs in this high-risk subgroup of patients. An essential finding in the present study was that the therapeutic effect of EPCs which were derived from severe PAOD patients prior to CD34^+^ cell and HBO therapy (i.e., non-rejuvenated EPCs) inadequately salvaged the nude mouse CLI, resulting in high incidence of automatic amputation and limb loss. Our findings, therefore, further support those of previous studies [21,22]. However, the results were inconsistent among the positive findings [16,17,18,19,20] and negative findings [21,22] of previous studies, and the findings of our study illustrated that some unidentified confounders might be present in these studies.

Undoubtedly, the fundamental pathological findings [8,10] demonstrate that the microvasculature of severe PAOD patients is always composed of severe thickness of intimal endothelial cell layer, a thickening and calcified medial layer and fibrosis of adventitia. Severe diffuse atherosclerotic changes in the arterioles and arteries, called “rusty vessels”, also accompany the critical limitation of distal run of blood flow that is always observed in these patients [8]. Additionally, the EPCs function is always found to be severely impaired in this high-risk population of severe PAOD patients [25]. These essential issues could explain why the EPCs can never mobilize into microvasculature and penetrate into the endothelial layer and home in on the ischemic zone for angiogenesis.

Studies have proposed that the mechanistic basis of HBO therapy for improving critical limb ischemia/severe PAOD patients was mainly through increased vascular wall permeability, enhanced productions of HIF-1α and SDF-1α and augmented diffusion of oxygen into the ischemic area that, in turn, enhanced the circulatory level of EPCs, angiogenesis and restoration of blood flow in the ischemic area [24,25]. The most important finding in the present study was that as compared with CLI and EPC^Pr-T^ groups, the INBF value (i.e., an indicator of restored blood flow in the ischemic area) was substantially increased, whereas the automatic amputation of the distal ischemic limb was remarkably reduced in the EPC^Af-T^ group. These findings highlight that rejuvenated EPCs may serve as an innovative therapy for CLI/severe PAOD patients, especially those patients who are refractory to conventional therapy.

A principal finding in the present study was that, when we looked at the cellular-molecular levels, we found that the angiogenesis factor, neovascularization and the integrity of endothelial cells were markedly increased in EPC^Af-T^ animals compared to those of CLI and CLI-EPC^Pr-T^ animals. On the other hand, the fibrosis and apoptosis biomarkers in the ischemic quadriceps were notably lower in the former group than in the latter two groups. These findings could explain why the number of distal ischemic limb losses was remarkably lower, whereas the INBF value was remarkably increased in EPC^Af-T^ animals compared to those of CLI and CLI-EPC^Pr-T^ animals.

### Study Limitation

This study contained limitations. First, in fact, the pathogenesis of the CLI model in nude mice was actually not identical to patient’s acute limb ischemia/severe PAOD. Accordingly, the underlying mechanisms of CLI between these two species could not be identical. Second, we did not completely exclude the presence or absence of the immune reaction after the patient’s EPCs were injected into the nude mice. Third, due to automatic amputation of distal ischemic limbs that occurred in the majority of CLI animals, we did not measure the INBF value by day 28 after the CLI procedure. Finally, although extensive works were done in the current study, the exactly mechanistic basis of xenogeneic EPC therapy for salvaging the nude mouse limbs from CLI is currently unclear. Perhaps, the proposed underlying mechanism in Figure 10 could, at least in part, provide useful information for further understanding this issue.

## 4. Materials and Methods

### 4.1. Ethics and Study Design

The EPCs utilized in this animal model of CLI were derived from circulatory mononuclear cells of severe PAOD patients who were treated with HBO and autologous CD34^+^ cells. This was a phase I clinical trial, which was approved by the Ministry of Health and Welfare, Taiwan, Republic of China (IRB No. 1076616554) and the Institutional Review Committee on Human Research at Chang Gung Memorial Hospital (IRB No. 201601217A0C602; date of approval 9 May 2018) in 2016. This phase I clinical trial was a prospective, randomized, open-label controlled trial to test the safety and efficacy of combined HBO and autologous circulatory CD34^+^ cell treatment for patients with severe PAOD at a single medical center. The occlusive level of the arteries was below the ankle, where catheter-based or surgical intervention was inappropriate and ineffective to treat the ischemic area.

All the patients agreed to receive this alternative treatment. A written informed consent form was obtained from every patient before enrollment and treatment.

This study was designed to consecutively enroll study patients who had received optimal medical care. The patients were enrolled to either receive CD34^+^ cells (2.5 × 10^7^) in the most severe PAOD of the lower extremity (group 1) or serve as control subjects with only standard pharmacotherapy (group 2) (i.e., 1:1 randomization, *n* = 10 in each group). This study remains active for the enrollment of patients.

### 4.2. Procedure and Protocol for Isolation of Autologous CD34^+^ Cells and Intra-Superficial Femoral Artery Infusion

The procedure and protocol for the isolation of circulatory CD34^+^ cells were based on our previous reports [18]. In detail, prior to the isolation of peripheral blood-derived CD34^+^ cells, granulocyte-colony stimulating factor (G-CSF) (5 μg/kg, every 12 h for eight doses) was subcutaneously given to each patient to augment the number of circulatory CD34^+^ cells for subsequent collection via leukapheresis. After the last dose of G-CSF, the mononuclear cells isolated during leukapheresis were enriched for CD34^+^ cells by using a commercially available device [COBE Spectra 6.1 (Terumo BCT Inc., Lakewood, CO, USA)] at 8:00 a.m. through a double-lumen catheter inserted into the right femoral vein. After a procedure time of about four hours, an adequate number of blood-derived CD34^+^ cells were isolated (purified through fluorescence-activated cell sorting for CD34^+^ cells) and ready for intra-renal artery transfusion.

After completing CD34^+^ cell collection, the patients were immediately sent to the cardiac catheterization room to receiving the intra-superficial femoral arterial transfusion of CD34^+^ cells into the target vessels below the ankle level.

### 4.3. Procedure and Protocol of Hyperbaric Oxygen Therapy

The procedure and protocol for hyperbaric oxygen therapy have been reported in our previous study [25]. The HBO therapy was performed for the patients in a sealed multi-place chamber at a pressure of 2.5 atmospheres absolute (ATA). Air pressure was gradually increased from 1 to 2.5 ATA over a 15 min duration. Oxygen of 100% medical-grade was inhaled through a plastic face mask for 25 min, followed by a 5 min break, for a total of 90 min per treatment. Air pressure was then decompressed from 2.5 ATA down to 1.0 ATA within 15 min to complete the treatment. HBO was performed daily, five times a week, for a total of 10–15 treatments.

All patients that received HBO therapy were safe without any complications, irrespective of recruitment from the hospitalization or outpatient department.

### 4.4. Ethics of Animal Model Study

All animal procedures were approved by the Institute of Animal Care and Use Committee at Kaohsiung Chang Gung Memorial Hospital (Affidavit of Approval of Animal Use Protocol No. 2017121301; date of approval 30 January 2018) and performed in accordance with the Guide for the Care and Use of Laboratory Animals.

Animals were housed in an Association for Assessment and Accreditation of Laboratory Animal Care International (AAALAC; Frederick, MD, USA)-approved animal facility in our hospital with controlled temperature and light cycles (24 °C and 12/12 light cycle).

### 4.5. Animal Model of CLI, Animal Grouping and Strategic Treatments

The procedure and protocol of CLI were based on our previous report [29]. Briefly, pathogen-free, adult male nude mice weighing 22–25 g (Charles River Technology, BioLASCO Taiwan Co., Ltd., Taiwan) in CLI groups were anesthetized by inhalation of 2.0% isoflurane. The nude mice were placed in a supine position on a warming pad at 37 °C with the left hind limbs shaved. Under sterile conditions, the left femoral artery, small arterioles, circumferential femoral artery and veins were exposed and ligated over their proximal and distal portions before removal. To avoid the presence of collateral circulation, the branches were removed altogether. For the laser Doppler study, 10 nude mice in each group were utilized and 6 nude mice in each group were used for cellular-molecular assessment. For animals that served as controls, only the arteries were isolated, without ligation.

For the purpose of the study, adult-male nude mice (*n* = 40) were equally categorized into group 1 (sham-operated control (i.e., SC)), group 2 (CLI), group 3 (CLI + EPCs, 6 × 10^5^ cells, derived from severe PAOD patient’s circulatory blood prior to CD34^+^ cell and HBO treatment (EPCPr-T) by intramuscular injection at 3 h after CLI induction), and group 4 (CLI + EPCs, 6 × 10^5^ cells, derived from severe PAOD patient’s circulatory blood after CD34^+^ cell and HBO treatment (EPC^Af-T^, i.e., defined as rejuvenated EPCs) by intramuscular injection at 3 h after CLI induction).

### 4.6. Peripheral Blood Collected and Cultured for Endothelial Progenitor Cells

Peripheral blood was collected from four patients (i.e., one for two nude mice in the first two patients and one for three nude mice in the latter two patients) at 8:00 a.m. after the 5th HBO therapy and 24 h after EPC therapy to severe PAOD patients. The procedure and protocol for EPC culture were based on our previous reports [29,30] with some modification. In brief, the blood sampling was performed on day 21 (i.e., day 0 was the date of the 5th iteration of HBO therapy) prior to nude mouse CLI induction for xenogeneic transfusion. Isolated mononuclear cells from peripheral blood were cultured in a 100 mm diameter dish with 10 mL DMEM culture medium containing 10% FBS for 21 days. Flow cytometric analysis was performed for the identification of cellular characteristics (i.e., EPC surface markers) after cell labeling with appropriate antibodies on day 21 of cell cultivation prior to implantation.

### 4.7. Measurement of Blood Flow with Laser Doppler

The procedure and protocol were based on our previous reports [29]. In brief, animals in each group were anesthetized by inhalation of isoflurane (2.0%) on day 2, 7, 14 after CLI induction. The animals were placed supine on a warming pad (37 °C), and blood flow was detected in both inguinal areas by a laser Doppler scanner (moorLDLS, Moor Instruments, Axminster, UK). This instrument scans an area of skin, which is evaluated by the distance between the mirror and the skin. The laser beam penetrates the normal tissue and part of the incident light that is scattered by moving red blood cells (RBCs) in microvasculature/small arterioles, forming a frequency broadening that is finally investigated by a photodetector. The RBC velocities and concentration give rise to Doppler frequency shifts and account for the strength of the signal. The Doppler shift is thus proportional to a blood-flow related variable and is exhibited as an arbitrary perfusion unit (PU). The mean blood flow is computed to yield an average number of pixels. Accordingly, the ratio of blood flow (BF) in the left (ischemic (I)) leg to right (normal (N)) leg (i.e., INBF) was computed, resulting in a ratio of PU. By day 28, the animals in each group were euthanized, and the quadriceps muscles were collected for individual study.

### 4.8. Vessel Density in CLI Area

The procedure and protocol for vessel density has been described in our previous report [29]. In detail, the IHC staining of small blood vessels was performed with α-SMA (1:400) as the primary antibody at room temperature for 1 h, followed by washing with PBS thrice. Ten minutes after the addition of anti-mouse HRP-conjugated secondary antibodies, the tissue sections were washed with PBS thrice. Then, 3,3’-diaminobenzidine (DAB) (0.7 gm/tablet) (Sigma-Aldrich) was added, followed by washing with PBS thrice after one minute. Finally, hematoxylin was added as a counterstain for nuclei, followed by washing twice with PBS after one minute. The angiogenesis was analyzed only in regions of demonstrably ischemic-regenerating quadriceps muscle. Three quadricep sections were analyzed in each nude mouse. To avoid bias, three selected HPFs (200×) were analyzed in each section. The mean number per HPF for each animal was then determined by the summation of all numbers divided by 9.

### 4.9. Western Blot Analysis

The procedure and protocol of western blot analysis was based on our previous reports [8,16,29,30]. In brief, equal amounts (30 µg) of protein extracts from ischemic quadriceps of the animals were loaded and separated by SDS-PAGE using 12% acrylamide gradients. The membranes were incubated with monoclonal antibodies against CD31 (1:1000, Abcam, Cambridge, UK), von Willebrand factor (vWF) (1:1000, Abcam), CXCR4 (1:1000, Abcam), VEGF (1:1000, Abcam), stromal cell-derived growth factor ((SDF)-1α) (1:1000, Cell Signaling), cytosolic cytochrome C (1:2000, BD), mitochondrial cytochrome C (1:1000, BD), endothelial nitric-oxide synthase (eNOS) (1:1000, Abcam), Bax (1:1000, Abcam), transforming growth factor ((TGF)-ß) (1:500, Abcam), phosphorylated (p)-Smad3 (1:1000, cell signaling), p-Smad1/5 (1:1000, cell signaling), bone morphogenetic protein (1:1000, Abcam), cleaved caspase 3 (c-Csp3) (1:1000, cell signaling), cleaved poly (ADP-ribose) polymerase (c-PARP) (1:1000, cell signaling) and hypoxic inducible factor ((HIF)-α) (1:1000, Abcam). Signals were detected with HRP-conjugated goat anti-mouse or goat anti-rabbit IgG. Proteins were transferred to nitrocellulose membranes, which were then incubated in the primary antibody solution (anti-DNP 1:150) for two hours, followed by incubation with a second antibody solution (1:300) for one hour at room temperature. The washing procedure was repeated eight times within 40 min. Immunoreactive bands were visualized by enhanced chemiluminescence (ECL; Amersham Biosciences, Little Chalfont, UK) and were then exposed to Biomax L film (Kodak, Rochester, NY, USA). For quantification, ECL signals were digitized using LabWorks software (UVP).

### 4.10. Immunohistochemical (IHC) and Immunofluorescent (IF) Staining

The procedure and protocols have been described by previous reports [8,16,29,30]. Briefly, sections were incubated with primary antibodies specifically against CD31 (1:200, BD Pharmingen, Franklin Lakes, NJ, USA), vWF (1:200, Abcam), CXCR4 (1:200, Abcam) and SDF-1α (1:100, Santa Cruz), while sections incubated with irrelevant antibodies served as controls. Three sections of kidney specimens from each rat were analyzed. For quantification, three random HPFs (400× for IHC and IF studies) were analyzed in each section. The mean number (expressed as a percentage) of positively-stained cells per HPF for each animal was first divided by the total positively DAPI-stained cells, then determined by the summation of all numbers divided by 9.

### 4.11. Statistical Analysis

Quantitative data were expressed as mean ± SD. Statistical analysis was adequately performed by ANOVA followed by the Bonferroni multiple comparison post hoc test. Statistical analysis was performed using SAS statistical software for Windows Version 8.2 (SAS Institute, Cary, NC, USA). A probability value of less than 0.05 was considered statistically significant.

## 5. Conclusions

In conclusion, our findings demonstrated that only rejuvenated EPC therapy could effectively restore the blood flow and salvage the critical ischemic limbs in the nude mouse model of CLI.

## Figures and Tables

**Figure 1 ijms-21-07887-f001:**
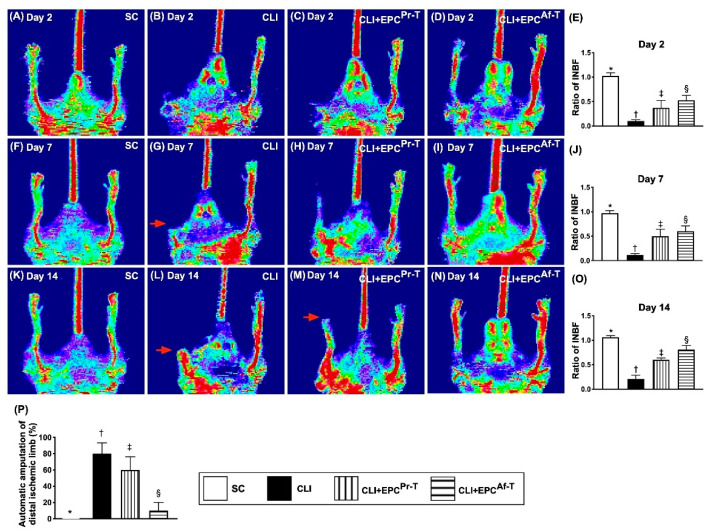
Ischemic/normal blood flow (INBF) ratio measured by laser Doppler scan at days 2, 7 and 14 after left femoral artery ligation. (**A**–**D**) Illustrating the laser Doppler finding of blood flow of right and left (CLI zone) limbs among the four groups at day 2 after CLI procedure; (**E**) Analytical result of ratio of INBF, * vs. other groups with different symbols (†, ‡, §), *p* < 0.0001; (**F**–**I**) Illustrating the laser Doppler finding of blood flow of right and left (CLI zone) limbs among the four groups at day 7 after CLI procedure; (**J**) Analytical result of ratio of INBF, * vs. other groups with different symbols (†, ‡, §), *p* < 0.0001; (**K**–**N**) Illustrating the laser Doppler finding of blood flow of right and left (CLI zone) limbs among the four groups at day 14 after CLI procedure; (**O**) Analytical result of ratio of INBF, * vs. other groups with different symbols (†, ‡, §), *p* < 0.0001; (**P**) Analytical result of percentage of automatic amputation of distal ischemic limb (red arrows) among the four groups by day 28 after CLI procedure, * vs. †, *p* < 0.0001. All statistical analyses are performed by one-way ANOVA, followed by the Bonferroni multiple comparison post hoc test (*n* = 10 for each group). Symbols (*, †, ‡, §) indicate significance (at 0.05 level). SC = sham-operated control; CLI = critical limb ischemia; EPCs = endothelial progenitor cells; EPC^Pr-T^ = EPCs derived from severe PAOD patient’s circulatory blood prior to CD34^+^ cell and HBO treatment; EPC^Af-T^ = EPCs derived from severe PAOD patient’s circulatory blood after CD34^+^ cell and HBO treatment; PAOD = peripheral arterial occlusive disease; HBO = hyperbaric oxygen.

**Figure 2 ijms-21-07887-f002:**
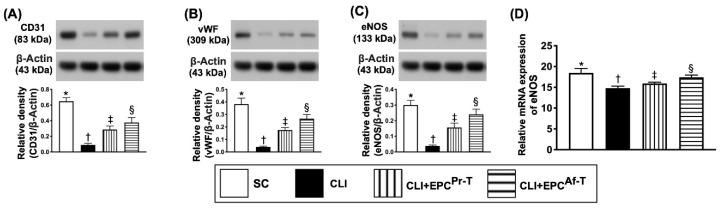
Protein expressions of endothelial cell functional integrity and gene expression of endothelial nitric-oxide synthase (eNOS) in CLI zone by day 28 after CLI procedure. (**A**) Protein expression of CD31, * vs. other groups with different symbols (†, ‡, §), *p* < 0.0001; (**B**) Protein expression of von Willebrand factor (vWF), * vs. other groups with different symbols (†, ‡, §), *p* < 0.0001; (**C**) Protein expression of eNOS, * vs. other groups with different symbols (†, ‡, §), *p* < 0.0001; (**D**) mRNA expression of eNOS, * vs. other groups with different symbols (†, ‡, §), *p* < 0.0001. All statistical analyses are performed by one-way ANOVA, followed by the Bonferroni multiple comparison post hoc test (*n* = 6 for each group). Symbols (*, †, ‡, §) indicate significance (at 0.05 level). SC = sham-operated control; CLI = critical limb ischemia; EPC = endothelial progenitor cells; EPC^Pr-T^ = EPCs derived from severe PAOD patient’s circulatory blood prior to CD34^+^ cell and HBO treatment; EPC^Af-T^ = EPCs derived from severe PAOD patient’s circulatory blood after CD34^+^ cell and HBO treatment; PAOD = peripheral arterial occlusive disease; HBO = hyperbaric oxygen.

**Figure 3 ijms-21-07887-f003:**
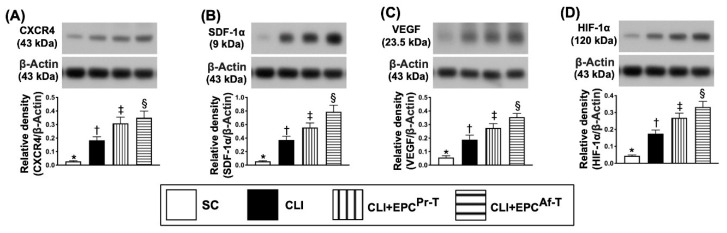
Protein expressions of angiogenesis factors in CLI zone by day 28 after CLI procedure. (**A**) Protein expression of CXCR4, * vs. other groups with different symbols (†, ‡, §), *p* < 0.0001; (**B**) Protein expression of stromal cell-derived factor (SDF)-1α, * vs. other groups with different symbols (†, ‡, §), *p* < 0.0001; (**C**) Protein expression of vascular endothelial growth factor (VEGF), * vs. other groups with different symbols (†, ‡, §), *p* < 0.0001; (**D**) Protein expression of hypoxia-inducible factor (HIF)-1α, * vs. other groups with different symbols (†, ‡, §), *p* < 0.0001. All statistical analyses are performed by one-way ANOVA, followed by the Bonferroni multiple comparison post hoc test (*n* = 6 for each group). Symbols (*, †, ‡, §) indicate significance (at 0.05 level). SC = sham-operated control; CLI = critical limb ischemia; EPC = endothelial progenitor cells; EPC^Pr-T^ = EPCs derived from severe PAOD patient’s circulatory blood prior to CD34^+^ cell and HBO treatment; EPC^Af-T^ = EPCs derived from severe PAOD patient’s circulatory blood after CD34^+^ cell and HBO treatment; PAOD = peripheral arterial occlusive disease; HBO = hyperbaric oxygen.

**Figure 4 ijms-21-07887-f004:**
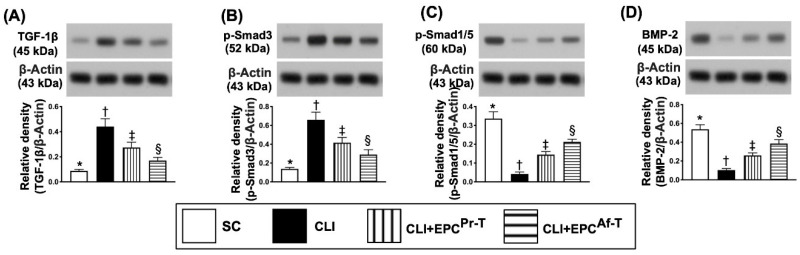
Protein expressions of fibrotic and antifibrotic biomarkers in CLI zone by day 28 after CLI procedure. (**A**) Protein expression of transforming growth factor (TGF)-ß, * vs. other groups with different symbols (†, ‡, §), *p* < 0.0001; (**B**) Protein expression of Smad3, * vs. other groups with different symbols (†, ‡, §), *p* < 0.0001; (**C**) Protein expression of Smad1/5, * vs. other groups with different symbols (†, ‡, §), *p* < 0.0001; (**D**) Protein expression of bone morphogenetic protein (BMP)-2, * vs. other groups with different symbols (†, ‡, §), *p* < 0.0001. All statistical analyses are performed by one-way ANOVA, followed by the Bonferroni multiple comparison post hoc test (*n* = 6 for each group). Symbols (*, †, ‡, §) indicate significance (at 0.05 level). SC = sham-operated control; CLI = critical limb ischemia; EPC = endothelial progenitor cells; EPC^Pr-T^ = EPCs derived from severe PAOD patient’s circulatory blood prior to CD34^+^ cell and HBO treatment; EPC^Af-T^ = EPCs derived from severe PAOD patient’s circulatory blood after CD34^+^ cell and HBO treatment; PAOD = peripheral arterial occlusive disease; HBO = hyperbaric oxygen.

**Figure 5 ijms-21-07887-f005:**
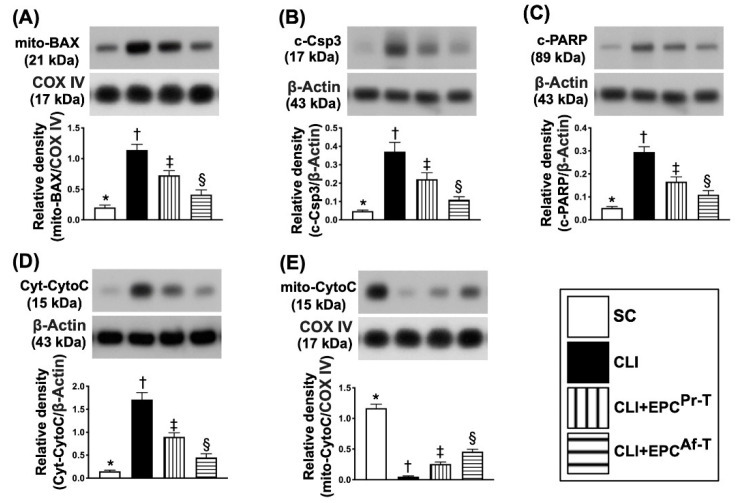
Protein expression of apoptotic, mitochondrial-damaged and mitochondrial-integrity biomarkers in CLI zone by day 28 after CLI procedure. (**A**) Protein expression of mitochondrial (mit-Bax), * vs. other groups with different symbols (†, ‡, §), *p* < 0.0001; (**B**) Protein expression of cleaved caspase 3 (c-Csp3), * vs. other groups with different symbols (†, ‡, §), *p* < 0.0001; (**C**) Protein expression of cleaved poly (ADP-ribose) polymerase 1 (c-PARP), * vs. other groups with different symbols (†, ‡, §), *p* < 0.0001; (**D**) Protein expression of cytosolic cytochrome C (cyt-Cyto-C), * vs. other groups with different symbols (†, ‡, §), *p* < 0.0001; (**E**) Protein expression of mitochondrial cytochrome C (mit-Cyto-C), * vs. other groups with different symbols (†, ‡, §), *p* < 0.0001. All statistical analyses are performed by one-way ANOVA, followed by the Bonferroni multiple comparison post hoc test (*n* = 6 for each group). Symbols (*, †, ‡, §) indicate significance (at 0.05 level). SC = sham-operated control; CLI = critical limb ischemia; EPC = endothelial progenitor cells; EPC^Pr-T^ = EPCs derived from severe PAOD patient’s circulatory blood prior to CD34^+^ cell and HBO treatment; EPC^Af-T^ = EPCs derived from severe PAOD patient’s circulatory blood after CD34^+^ cell and HBO treatment; PAOD = peripheral arterial occlusive disease; HBO = hyperbaric oxygen.

**Figure 6 ijms-21-07887-f006:**
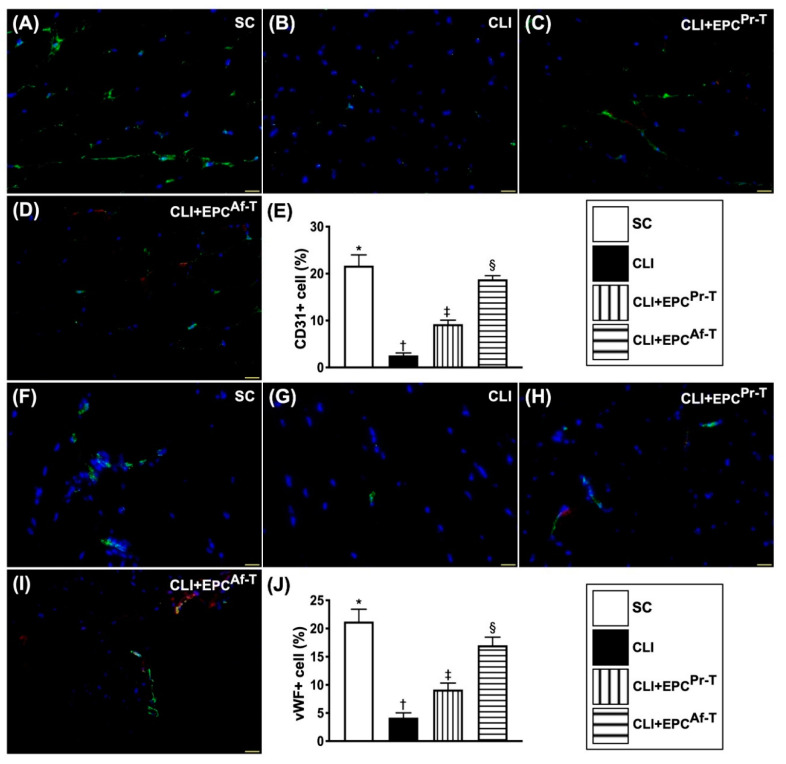
Cellular expressions of endothelial cell surface markers in CLI zone by day 28 after CLI procedure. (**A**–**D**) Illustrating the immunofluorescent (IF) microscopic finding (400×) for identification of CD31^+^ cells (green color) in the CLI area; (**E**) Analytical result of number of CD31+ cells, * vs. other groups with different symbols (†, ‡, §), *p* < 0.0001; (**F**–**I**) Illustrating the IF microscopic finding (400×) for identification of von Willebrand factor (vWF)^+^ cells (green color) in the CLI area; (**J**) Analytical result of the number of vWF+ cells, * vs. other groups with different symbols (†, ‡, §), *p* < 0.0001. Scale bars in right lower corner represent 20 µm. All statistical analyses are performed by one-way ANOVA, followed by Bonferroni multiple comparison post hoc testing (*n* = 6 for each group). Symbols (*, †, ‡, §) indicate significance (at 0.05 level). SC = sham-operated control; CLI = critical limb ischemia; EPC = endothelial progenitor cells; EPC^Pr-T^ = EPCs derived from severe PAOD patient’s circulatory blood prior to CD34^+^ cell and HBO treatment; EPC^Af-T^ = EPCs derived from severe PAOD patient’s circulatory blood after CD34^+^ cell and HBO treatment; PAOD = peripheral arterial occlusive disease; HBO = hyperbaric oxygen.

**Figure 7 ijms-21-07887-f007:**
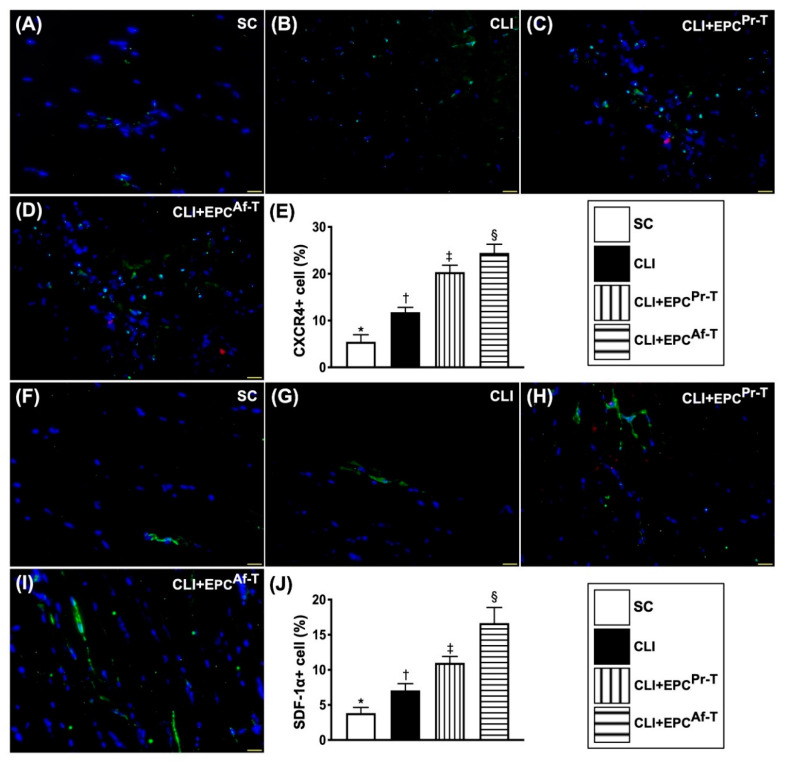
Cellular expressions of angiogenesis markers in CLI zone by day 28 after CLI procedure. (**A**–**D**) Illustrating the immunofluorescent (IF) microscopic finding (400×) for identification of CXCR4^+^ cells (green color) in the CLI area; (**E**) Analytical result of number of CXCR4^+^ cells, * vs. other groups with different symbols (†, ‡, §), *p* < 0.0001; (**F**–**I**) Illustrating the IF microscopic finding (400×) for identification of stromal cell-derived factor (SDF)-1α^+^ cells (green color) in CLI area; (**J**) Analytical result of number of SDF-1α^+^ cells, * vs. other groups with different symbols (†, ‡, §), *p* < 0.0001. All statistical analyses are performed by one-way ANOVA, followed by the Bonferroni multiple comparison post hoc test (*n* = 6 for each group). Symbols (*, †, ‡, §) indicate significance (at 0.05 level). SC = sham-operated control; CLI = critical limb ischemia; EPC = endothelial progenitor cells; EPC^Pr-T^ = EPCs derived from severe PAOD patient’s circulatory blood prior to CD34^+^ cell and HBO treatment; EPC^Af-T^ = EPCs derived from severe PAOD patient’s circulatory blood after CD34^+^ cell and HBO treatment; PAOD = peripheral arterial occlusive disease; HBO = hyperbaric oxygen.

**Figure 8 ijms-21-07887-f008:**
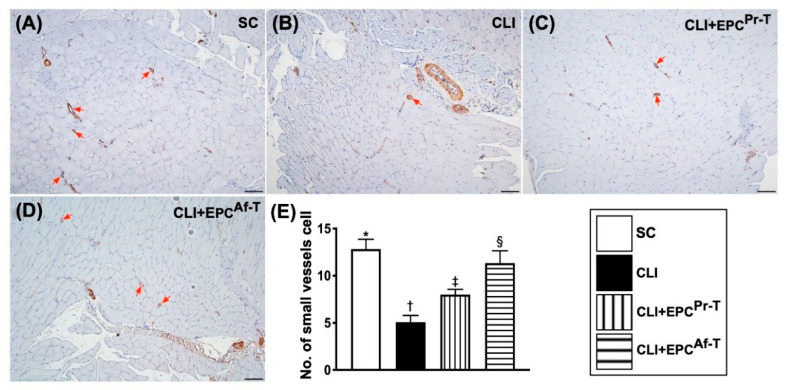
Small vessel density in CLI zone by day 28 after CLI procedure. (**A**–**D**) Illustrating the microscopic findings (100×) of alpha-smooth muscle actin (α-SMA) for identification of the number of small vessels (i.e., diameter ≤ 25.0 μM) (red arrows). (**E**) Analytic result of the number of small vessels, * vs. other groups with different symbols (†, ‡, §), *p* < 0.0001. All statistical analyses are performed by one-way ANOVA, followed by Bonferroni multiple comparison post hoc testing (*n* = 6 for each group). Symbols (*, †, ‡, §) indicate significance (at 0.05 level). SC = sham-operated control; CLI = critical limb ischemia; EPC = endothelial progenitor cells; EPC^Pr-T^ = EPCs derived from severe PAOD patient’s circulatory blood prior to CD34^+^ cell and HBO treatment; EPC^Af-T^ = EPCs derived from severe PAOD patient’s circulatory blood after CD34^+^ cell and HBO treatment; PAOD = peripheral arterial occlusive disease; HBO = hyperbaric oxygen.

**Figure 9 ijms-21-07887-f009:**
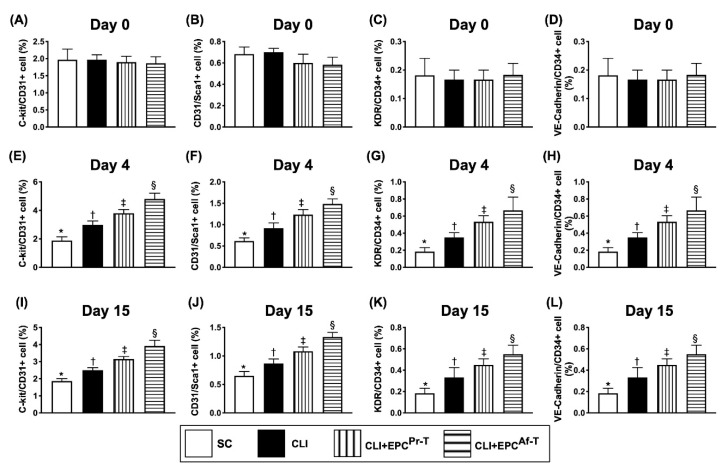
Illustrating the time courses of flow cytometric analysis of circulating levels of EPCs. (**A**–**D**) By day 0, the analytical of numbers of circulating levels of c-kit/CD31^+^ (**A**), CD31/sca-1^+^ (**B**), KDR/CD34^+^ (**C**) and VE-Cadherin/CD34^+^ (**D**) cells, all *p* value > 0.5; (**E**–**H**) By day 4, the analytical of numbers of circulating levels of c-kit/CD31^+^ (**E**), CD31/sca-1^+^ (**F**), KDR/CD34^+^ (**G**) and VE-Cadherin/CD34^+^ (**H**) cells, all *p* value < 0.0001; (**I**–**L**) By day 15, analytical of numbers of circulating levels of c-kit/CD31^+^ (**I**), CD31/sca-1^+^ (**J**), KDR/CD34^+^ (**K**) and VE-Cadherin/CD34^+^ (**L**) cells, all *p* value < 0.0001. All statistical analyses are performed by one-way ANOVA, followed by the Bonferroni multiple comparison post hoc test (*n* = 6 for each group). Symbols (*, †, ‡, §) indicate significance (at 0.05 level). SC = sham-operated control; CLI = critical limb ischemia; EPC = endothelial progenitor cells; EPC^Pr-T^ = EPCs derived from severe PAOD patient’s circulatory blood prior to CD34^+^ cell and HBO treatment; EPC^Af-T^ = EPCs derived from severe PAOD patient’s circulatory blood after CD34^+^ cell and HBO treatment; PAOD = peripheral arterial occlusive disease; HBO = hyperbaric oxygen.

**Figure 10 ijms-21-07887-f010:**
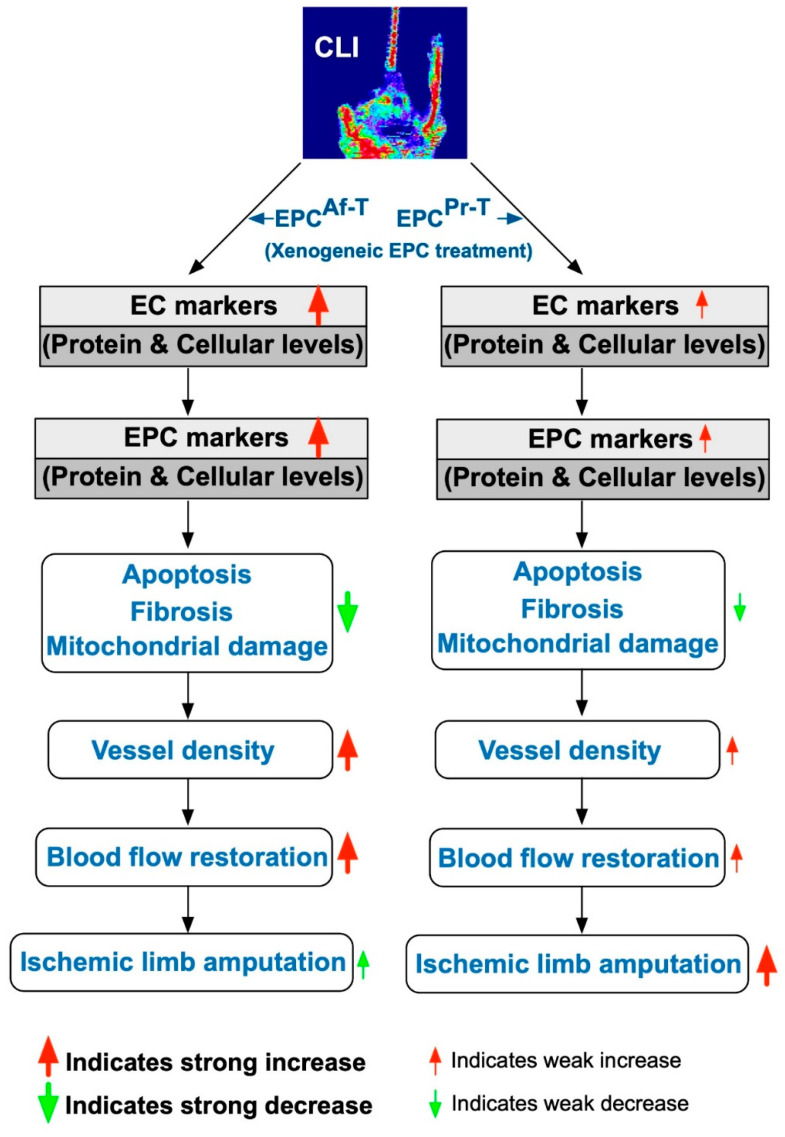
Illustrating the proposed underlying mechanism for rejuvenated xenogeneic EPCs effectively salvaging nude mouse limbs from CLI. CLI = critical limb ischemia; EPC = endothelial progenitor cells; EPC^Pr-T^ = EPCs derived from severe PAOD patient’s circulatory blood prior to CD34^+^ cell and HBO treatment; EPC^Af-T^ = EPCs derived from severe PAOD patient’s circulatory blood after CD34^+^ cell and HBO treatment.

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
