# Peer review of "Circulatory Rejuvenated EPCs Derived from PAOD Patients Treated by CD34+ Cells and Hyperbaric Oxygen Therapy Salvaged the Nude Mouse Limb against Critical Ischemia"

_ijms, 2020, doi:10.3390/ijms21217887_

Round 1

Reviewer 1 Report

In the current study, the authors report ischemic-protecting properties of circulatory endothelial progenitor cells (EPC) isolated from peripheral arterial disease patients who received autologous CD34+ cells hyperbaric oxygen therapy. They have shown ischemic-protecting properties of these treated human EPCs superior to those without any treatment in an experimental mouse model of hindlimb ischemia. However, there are some concerns and questions about the experimental approach and design are outlined below:

  • Did the therapeutic combination of hyperbaric therapy and CD34+ cell therapy improve the clinical outcome of PAD patients in the tested clinical study? In the current paper, this information is missing. The authors are advised to include this information in the current paper.
  • The study lacks flow cytometry studies that provide a characterization of isolated EPC. The authors should add these characterization studies.
  • It is not clear the rationale behind selecting the quadriceps muscles for the subsequent histological and biochemical analysis. It is uncommon to analyze this muscle group for ischemic analysis. The calf muscles are the most common site to study.
  • Methods, p14, line 417, “peripheral blood was collected from four patients (i.e., One for two nude mice.” It means that five different EPC’s patient samples were only tested in each group. The data of each two mice that received EPC from one patient should be averaged. The authors should re-evaluate their blood flow measurement results and histological studies based on five treated mice per group and not as ten treated mice per group.
  • Methods, p14, line 441: “For quantification, three randomly selected HPFs (200x) were analyzed in each section.” The muscle injury extent is usually not uniform all over the ischemic muscles, and angiogenesis is also patchy and exclusively associated with regenerated muscle zones. The angiogenesis should not be quantified randomly, and instead, angiogenesis should be analyzed only in regions of demonstrably regenerating muscle.
  • The authors have reported most of the histological quantification as a percentage. It is not clear how this percentage was calculated. The authors should clarify the calculation method. Also, vascular density is reported as a percentage. The vascular density (angiogenesis) should be calculated as a capillary-to-fiber ratio or capillary number per muscle area.

Minor points:

  • Figure 9. the arrows are confusing. For instance, at the right panel, “ischemic limb amputation” indicates a substantial increase in the animal group received EPCAf-T. This conclusion seems to contradict the findings of the study. The authors should simplify this figure.
  • The authors are advised to show representative histological images with lower magnification.

Author Response

Point 1: Did the therapeutic combination of hyperbaric therapy and CD34+ cell therapy improve the clinical outcome of PAD patients in the tested clinical study? In the current paper, this information is missing. The authors are advised to include this information in the current paper.

Response 1: First of all, thank you for your professional comment. However, we deeply apology for that we can’t provide this clinical information for several reasons: first, the Taiwan FDA (i.e., the Ministry of Health and Welfare, Taiwan) and our hospital IRB strongly recommend that we are not permitted to publish the clinical data when the clinical trial is still on going , especially when the efficacy is discussed for an ongoing clinical trial (just as we had mentioned in Mthedology Section “4.1. Ethics and study design” that the “This study remains active for enrollment of patients.”. Second, due to the phase I clinical trial is prospectively randomized and at the present time only 7 patients (i.e., 3 control and 4 in treatment group) have been enrolled. Although some parameters and questionnaire support a favorable clinical outcome in treatment group, to analyze such a small sample size would really distort the statistical significance that surely can’t convince the readers. Accordingly, we did not provide this information and whish that you can understand our inconvenience. (in red)

Point 2: The study lacks flow cytometry studies that provide a characterization of isolated EPC. The authors should add these characterization studies.

Response 2: Yes, we have provided this information in Figure 9 (i.e., new Figure) in Result Section and Figure Legend of our revised manuscript. (in red)

Point 3: It is not clear the rationale behind selecting the quadriceps muscles for the subsequent histological and biochemical analysis. It is uncommon to analyze this muscle group for ischemic analysis. The calf muscles are the most common site to study.

Response 3: Dear reviewer, in the present study, we selected the quadriceps muscles for individual analysis due to (1) the quadriceps muscle in situation of ischemia that was suitable for the purpose of the study and (2) the majority of limb was automatic amputation in the CLI, resulting in loss of the calf muscles. Accordingly, we could not get the calf muscles for molecular-cellular study. (in red)

Point 4: Methods, p14, line 417, “peripheral blood was collected from four patients (i.e., One for two nude mice.” It means that five different EPC’s patient samples were only tested in each group. The data of each two mice that received EPC from one patient should be averaged. The authors should re-evaluate their blood flow measurement results and histological studies based on five treated mice per group and not as ten treated mice per group.

Response 4: Dear reviewer, we sincerely apology for our inappropriate writing. Correctly, the sentence should be written as “i.e., one for two nude mice in first two patients and one for three mice in the later two patients”. Finally, the analysis was combined together for each parameter. This has been rewritten in Method Section “4.6. Peripheral blood collected and cultured for endothelial progenitor cells” of our revised manuscript. (in red). Really, we want to to comply with your honorable suggestions. However, as you know that we can’t repeat to collect the blood samples in the same patients because the ethical issue.   

Point 5: Methods, p14, line 441: “For quantification, three randomly selected HPFs (200x) were analyzed in each section.” The muscle injury extent is usually not uniform all over the ischemic muscles, and angiogenesis is also patchy and exclusively associated with regenerated muscle zones. The angiogenesis should not be quantified randomly, and instead, angiogenesis should be analyzed only in regions of demonstrably regenerating muscle.

Response 5: Dear reviewer, we are honest to tell you that the angiognesis was really analyzed in the ischemic quadriceps muscle. We have stated more accurately in Mehtodology Section “4.8. Vessel density in CLI area” of our revised manuscript. (in red)

Point 6: The authors have reported most of the histological quantification as a percentage. It is not clear how this percentage was calculated. The authors should clarify the calculation method. Also, vascular density is reported as a percentage. The vascular density (angiogenesis) should be calculated as a capillary-to-fiber ratio or capillary number per muscle area.

Response 6: (1) For the small vessel density, we calculated the number of small vessels in a high-power field. This method has been validated by previous reported study (Crit Care Med 2012, 40, (1), 169-77.). According to your honorable comment, we have cited this reference in the Methodology Section “4.8. Vessel density in CLI area” of our revised manuscript. (2) The method for the histological quantification as a percentage has been clarified for more detail in Method Section “4.10. Immunohistochemical (IHC) and immunofluorescent (IF) staining” of our revised manuscript. (in red)

Response to minor comment

Point 1: Figure 9. the arrows are confusing. For instance, at the right panel, “ischemic limb amputation” indicates a substantial increase in the animal group received EPCAf-T. This conclusion seems to contradict the findings of the study. The authors should simplify this figure.

Response 1: We apology for our mistake. This mistake has been corrected in our new Figure 10 (it was old Figure 9). (in red)

Point 2: The authors are advised to show representative histological images with lower magnification.

Response 2: Dear reviewer, your constructive criticism is greatly appreciated. In Figure 8, the magnification of histological image is 100x (i.e., relatively lower). In Figures 6 and 7, we had tried the magnification of histological image to be 200x in our original microscopic findings. However, the images were concealed and not so clear. This is why utilized the 400x magnification in the histological images. We hope that you can understand this inconvenience. (in red)        

We are greatly indebted to you for your professional comments and suggestions!

Reviewer 2 Report

An interesting study supporting the issue of reduced functionality of aging EPC:s in CLI patients. Young cells are supposed to be more effective both regarding angiogenesis and modification of immunemodulation.

Most importantly -as also discussed by the authors--positive effects on mice does not prove any effect in humans. 

Just two issues for clarification

  1. How was a "CLI" condition created in the mice ? One main artery ligature or complete "skeletonization" 
  2. INBF should be better explained (calculation) 

Author Response

Response to reviewer 2 comments

Point 1: How was a "CLI" condition created in the mice ? One main artery ligature or complete "skeletonization"

Response 1: Dear reviewer, in the “Section of 4.5. Animal model of CLI, animal grouping and strategic treatments”, we have stated that “the left femoral artery, small arterioles, circumferential femoral artery and veins were exposed and ligated over their proximal and distal portions before removal. To avoid the presence of collateral circulation, the branches were removed altogether.” Suggesting that this mehtod colud be called complete "skeletonization". (in pink)

Point 2: INBF should be better explained (calculation)

Response 2: Yes, according to you recommendatation, we have provided this infromation in Methodology Section “4.7. Measurement of blood flow with laser Doppler” of our revised manuswcript. (in pink)

We would like to take this opportunity to express our appreciation for your detailed review of the article and the kindness of giving us valuable suggestions. Thank you very, very much!

Round 2

Reviewer 2 Report

Corrections according to my comments are appropriate

Author Response

Response to reviewer 2 comments

Point 1: Corrections according to my comments are appropriate

Response 1: We would like to thank reviewer for the insightful comments and the time taken to review our manuscript. In this way, we have carefully read our manuscript and tried our best to correct the mistakes again. Please kindly let us know if there is further need for improving our work.

We would like to take this opportunity to express our appreciation for your detailed review of the article and the kindness of giving us valuable suggestions. Thank you very, very much!